# Biological Predictors of Osteoarticular Infection Due to *K. kingae*—A Retrospective Cohort Study of 247 Cases

**DOI:** 10.3390/microorganisms11092130

**Published:** 2023-08-22

**Authors:** Blaise Cochard, Giacomo De Marco, Ludmilla Bazin, Oscar Vazquez, Giorgio Di Laura Frattura, Christina N. Steiger, Romain Dayer, Dimitri Ceroni

**Affiliations:** 1Pediatric Orthopedics Unit, Pediatric Surgery Service, Geneva University Hospitals, CH-1211 Geneva, Switzerland; blaise.cochard@hcuge.ch (B.C.); ludmilla.bazin@hcuge.ch (L.B.); oscar.vazquez@hcuge.ch (O.V.); giorgio.dilaurafrattura@hcuge.ch (G.D.L.F.); romain.dayer@hcuge.ch (R.D.); 2Division of Orthopedics and Trauma Surgery, Geneva University Hospitals, CH-1211 Geneva, Switzerland

**Keywords:** *Kingella kingae*, osteoarticular infection, children

## Abstract

Pediatric osteoarticular infections (OAIs) are serious conditions that can lead to severe septic complications, prolonged morbidity with long-term impaired function, and perturbed subsequent bone development. *Kingella kingae* (*K. kingae*) is currently accepted as the predominant pathogen in pediatric OAIs, especially among 6–48 month olds. The present study aimed to identify clinical and biological markers that would refine the detection of patients with an OAI due to *K. kingae*. We retrospectively studied every consecutive case of pediatric OAI admitted to our institution over 17 years. Medical records were examined for patient characteristics such as temperature at admission, affected segment, and biological parameters such as white blood cell (WBC) count, left shift, platelet count (PLT), C-reactive protein (CRP), and erythrocyte sedimentation rate (ESR). The 247 patients included 52.2% males and 47.8% females and mean age was 18.5 ± 10 months old. Four patients were older than 48 months; none were younger than 6 months old. Mean temperature at admission was 37.4 ± 0.9 °C. Regarding biological parameters, mean WBC count was 12,700 ± 4180/mm^3^, left shift was only present in one patient, mean PLT was 419,000 ± 123,000/mm^3^, mean CRP was 26.6 ± 27.8 mg/L, and mean ESR was 35.0 ± 18.9 mm/h. Compared to the modified predictors of OAI defined by Kocher and Caird, 17.2% of our cases were above their cut-off values for temperature, 52.3% were above the WBC cut-off, 33.5% were above the ESR cut-off, and 46.4% were above the CRP cut-off. OAIs due to K. kingae frequently remain undetected using the classic biological parameters for investigating bacterial infections. As an addition to the predictors normally used (°C, WBC, CRP, and ESR), this study found that elevated platelet count was frequently present during OAIs caused by *K. kingae*. Although this biological characteristic was inconstant, its presence was highly significant and very suggestive of an invasive infection due to *K. kingae*.

## 1. Introduction

Osteoarticular infections (OAIs) among pediatric populations are considered serious conditions that can lead to severe septic complications, induce prolonged morbidity, perturb subsequent bone development, and cause long-term functional disability [1]. Better recognition of the microbiological causes of OAIs has evolved drastically since the early 2000s, and it is currently accepted that their clinical and biological characteristics are closely related to the child’s age and the incriminated pathogen. Indeed, some of these factors, e.g., associated comorbidities, immune and vaccination status, changes in the patterns of immunomodulating diseases, the emergence of resistant bacteria, and even the patient’s socioeconomic situation, now constitute such crucial parameters that not considering them is impossible [2].

For decades, *Staphylococcus aureus* (S. aureus) was considered the most prevalent pathogen in pediatric OAIs [3,4,5,6,7,8]. This biased view of the epidemiology probably stalled research on bacterial liability and new developments in diagnosing and treating OAIs. Since the early 2000s, however, cases of OAIs attributed to *Kingella kingae* (K. kingae) have increased drastically thanks to the widespread use of nucleic acid amplification assays. This new diagnostic approach has provided irrefutable evidence that *K. kingae* has become the most common pathogen guilty for primary infections in bones, joints, intervertebral discs, and tendon sheaths, especially among children aged 6–48 months old [9,10,11,12].

Interestingly, a *K. kingae* OAI is often characterized by a mild clinical presentation and a limited biological inflammatory response, with the consequence that young patients present few, if any, of the normal criteria evocative of an OAI. Contrarily, pyogenic OAIs caused by more aggressive pathogens, e.g., methicillin-sensitive or methicillin-resistant *S. aureus* (MSSA and MRSA), streptococci, or Gram-negative organisms, are more obvious, with affected children usually appearing quite ill and presenting with high fever and an elevated white blood cell (WBC) count [13,14].

Evaluating the clinical and biological parameters normally used (i.e., temperature, WBC count, C-reactive protein (CRP), and erythrocyte sedimentation rate (ESR)) to confirm or refute an OAI occurs within the framework of our knowledge about how they vary during infections due to pyogenic germs. Interpreting these biological parameters using the normative cut-off values is questionable during OAIs caused by *K. kingae* because they are not appropriate for the characteristics of an infection due to that microorganism. On this point, some authors have unexpectedly noted that children with a *K. kingae* OAI had a mean platelet count greater than children whose infections were caused by pyogenic microorganisms [9,10,11]. Showing a statistically significant difference in platelet counts between patients with *S. aureus* and *K. kingae* OAIs (*p* < 0.001), platelet counts were incorporated into a predictive model to differentiate between these two types of OAIs [15]. The present study aimed, therefore, to investigate and describe the clinical presentation and biological inflammatory response to *K. kingae* OAIs in a large cohort of patients and then define the role that platelet count might play in recognizing OAIs caused by this specific pathogen. This paper also attempts to summarize and critically examine the current evidence for OAIs due to K. kingae and focuses specifically on how well clinical and biological laboratory diagnostics can help us to recognize infections.

## 2. Materials and Methods

After approval by the Children’s Hospital Ethics Review Committee (CE 2023-102R), we retrospectively reviewed the medical charts of all children admitted to our institution, between January 2007 and May 2023, for OAIs caused by *K. kingae* (January 2007 corresponds to the start of the routine implementation of molecular detection methods for *K. kingae*). Our 111-bed tertiary pediatric hospital serves the city of Geneva and surrounding areas and is the only institution providing inpatient and specialized medical services for pediatric OAIs to 460,000 people. The diagnostic codes for septic arthritis, osteomyelitis (acute and subacute), spondylodiscitis, pyomyositis, and tenosynovitis were used to identify the study population in our institution’s electronic medical records.

Children’s risks of having a bone or joint infection were estimated using the criteria established by Morrey [16,17,18] and Morrissey [19]. Diagnoses of an OAI were confirmed using magnetic resonance imaging according to established criteria [20]. Children diagnosed with a musculoskeletal infection were further categorized using the following diagnoses: septic arthritis, osteomyelitis with concomitant septic arthritis, osteomyelitis, spondylodiscitis, septic myositis, septic chondritis, and septic tenosynovitis.

We collected and analyzed data on patient age, sex, temperature at admission, the bone or joint involved, and laboratory results for bacterial cultures (blood, synovial fluid, and bone exudate), quantitative polymerase chain reaction (qPCR) assays [21], WBC and differential platelet counts, ESR, and serum CRP. We used the following four classic cut-off values considered to have predictive value as infection parameters in clinical practice: fever defined as an oral temperature of ≥38 °C; WBC > 17,500/mm^3^ in children younger than 12 months old, >17,000/mm^3^ in children 13–24 months old, >14,500/mm^3^ in children 25–48 months old, and >12,000/mm^3^ in older children; CRP > 10 mg/L; and an ESR > 20 mm/h. We also tried to evaluate our results with regards to the cut-offs for the four main clinical and biological predictors for OAIs described in the Kocher/Caird algorithms, i.e., fever > 38.5°, WBC > 12.000/mm^3^, ESR ≥ 40 mm/h, and CRP ≥ 20 mg/L [22,23,24,25]. An elevated platelet count is currently not recognized as a diagnostic marker of OAI, even if a platelet count > 361,500/mm^3^ has now been integrated into a model for differentiating *K. kingae* OAIs from those caused by MSSA [9]. Platelet count, therefore, was evaluated using the absolute value beyond which the patient is considered to have thrombocytosis (>392,000/mm^3^) while also considering the threshold limit (>361,500/mm^3^) suggestive of an OAI caused by *K. kingae.*

Confirmed cases of OAI due to *K. kingae* were evidenced by positive imaging studies (standard radiography and MRI) and the pathogen’s successful isolation from blood, bone, and/or joint fluid cultures or PCR assays. We considered that children’s OAIs were highly probably due to *K. kingae* when clinical presentations and imaging studies were positive and they aligned with oropharyngeal specimen PCR assays also positive for *K. kingae*. Patients presumed to have an OAI due to *K. kingae*—who had positive imaging studies but whose clinical and laboratory data were only suggestive of the microorganism—were excluded from the study. Study exclusion criteria also included chronic osteomyelitis and infections subsequent to a fracture or surgery. Additive exclusion criteria were also used to avoid any information bias associated with an incomplete data analysis and selection bias associated with the inclusion of patients with presumptive and inconsistent diagnoses. These criteria were (i) no bacteriological diagnosis, (ii) no available laboratory data, and (iii) the patient was not eventually managed by administering antibiotics.

### 2.1. Microbiological Methods

Blood cultures have been used systematically to attempt to isolate the microorganisms responsible for OAIs. The present study used BACTEC 9000 blood culture media before 2009 and then the BD BACTEC FX automated blood culture system. Joint fluid or bone aspirates were sent to the laboratory for Gram staining, cell count, and immediate inoculation onto Columbia blood agar (incubated under anaerobic conditions), CDC anaerobe 5% sheep blood agar (incubated under anaerobic conditions), chocolate agar (incubated in a CO_2_-enriched atmosphere), and brain–heart broth. These media were incubated for 10 days. Two PCR assays were also used for bacterial identification when standard cultures were negative. Initial aliquots (100–200 µL) were stored at −80 °C until processing for DNA extraction. A universal, broad-range PCR amplification of the 16S rRNA gene was performed using BAK11w, BAK2, and BAK533r primers (Eurogentec, Seraing, Belgium). As of 2007, we also used a real-time PCR assay targeting the *K. kingae* gene’s rtx toxin [21]. This assay is designed to detect two independent gene targets from the *K. kingae* rtx toxin locus, namely *rtxA* and *rtxB* [21]; it was used to analyze different biological samples, such as synovial fluid, bone, or discal biopsy specimens, or peripheral blood. Since September 2009, our institution has also been performing oropharyngeal swab PCRs for children from 6 months to 4 years old. This has been shown to be a simple technique for detecting *K. kingae* rtx toxin genes in the oropharynx and provides strong evidence that this microorganism is responsible for an OAI or even stronger evidence that it is not [26].

### 2.2. Biological Methods

WBC and PLT counts were performed on Sysmex XN-1000 analyser (Sysmex Corporation, Kobe, Japan). Determination of WBC counts is based on flow cytometry method where laser light scattering technology is used. PLT counts are determined using the impedance method with hydrodynamic focusing. CRP measurement is determined using the immunoturbidimetry on Cobas e702 analyser (Roche Diagnostics GmBH, Mannheim, Germany). ESR is determined on Alifax Roller 20 PN (Sysmex Corporation, Kobe, Japan).

### 2.3. Statistical Analysis

The characteristics of patients with an OAI caused by *K. kingae* were analyzed, and clinical manifestations and laboratory test results were expressed as medians and ranges, as well as means and standard deviations. All statistical analyses were performed using Jamovi software, version 2.3 (The Jamovi project (2022), accessed from https://www.jamovi.org on 25 May 2023).

## 3. Results

The medical files of 497 children with a diagnostic of OAI have been analyzed. Among them, 313 patients were aged between 6 and 48 months. In total, 247 children with OAIs were attributed to *K. kingae* and were finally included in this study. The microorganism’s causal responsibility was proven in 167 cases (67.6%) and, by the remaining 80 cases (32.4%), infections caused by *K. kingae* were considered as highly probable. Slightly more male children (52.2%) were affected by OAIs due to *K. kingae*, and mean age (±SD) was 18.5 ± 10 months old, ranging from 6 to 79 months. There were no confirmed *K. kingae* OAIs in children under 6 months old, whereas there were four confirmed cases in children above 4 years old. The types and locations of the OAIs are listed in Table 1 and Table 2. *K. kingae* mostly caused septic arthritis of big joints and primarily affected knees and hips. *K. kingae* was identified using blood PCR in 5.6% of cases, using bone/fluid culture in 5.9% of cases, using a bone/joint fluid PCR assay specific for *K. kingae* in 87.3% of cases, and via positive PCR assays using oropharyngeal swabs in 95.5%.

The distribution of clinical and laboratory parameters is summarized in Table 3. We observed that 71.7% of patients with a *K. kingae* OAI were afebrile (T < 38 °C) at their clinical admission examination, and only 17.2% of them had a temperature superior to 38.5 °C. The mean WBC value was 12,700 ± 4,180/mm^3^; WBC count was considered elevated, accordingly to their normal age-related cut-offs, in 16.0% (>17,500/mm^3^ in children 6–12 months old, >17,000/mm^3^ in children 13–24 months old, >14,500/mm^3^ in children 24–48 months old, and >12,000/mm^3^ in older children) of the patients, whereas left shift was only noted in one patient with concomitant herpangina. On the other hand, WBC counts were superior to the threshold of >12′000/mm3 described by Kocher et al. [22] in 52.3% of children. The mean CRP value was 26.6 ± 27.8 mg/L, with CRP considered abnormal (>10 mg/L) in 64.1% of patients. Looking at the CRP value considered a predictor of a *K. kingae* OAI by Caird et al. [25] (CRP ≥ 20 mg/L), we noted that only 46.4% of patients with an OAI caused by *K. kingae* had a value above this limit. The mean ESR value reached 35.0 ± 18.9 mm/h and was considered abnormal (>20 mm/h) in 75.5% of patients. Looking at the ESR value considered a predictor of septic arthritis by Kocher et al. (ESR ≥ 40 mm/h), we noted that only 33.5% of patients with an OAI caused by *K. kingae* had a rate above this limit. Finally, the mean of the platelet count value was 419,000/mm^3^ ± 123,000/mm^3^. Platelet count values were considered elevated (>392,000/mm^3^) in 53.7% of cases (Table 4) but, when we applied the threshold limit (>361,500/mm^3^) suggestive of an OAI caused by *K. kingae* [9], we noted that 64.2% of our patients had values considered abnormal.

## 4. Discussion

Since the early 2000s, the increasing use of nucleic acid amplification assays in diagnostic processes has changed the generally recognized bacteriological epidemiology of OAIs [9,10,11], and *K. kingae* and MSSA are currently considered the two main pathogens responsible for them [9,10,11,27]. In view of their very different clinical courses, being able to discriminate between them rapidly seems essential [28], and there have been many attempts to build algorithms to distinguish between OAIs caused by *K. kingae* and those due to pyogenic pathogens [9,29]. There is also a current trend towards using traditional algorithms for quickly detecting all musculoskeletal infections, regardless of their type and location [30]. Conveniently, we also noted that, in previous studies, reactive thrombocytosis was more often associated with *K. kingae* OAIs than with MSSA infections [9,10,11]. The present work represents the largest consecutive case series that paid very specific attention to temperature and inflammatory biological responses during OAIs caused by *K. kingae* and which was interested in defining platelet count as a new marker of OAI.

Firstly, this study corroborated existing evidence indicating that most OAIs caused by *K. kingae* occur in children younger than 4 years old. Indeed, only three cases in the series deviated from this rule. It is important to remember that this specific time in a child’s life corresponds to the period of maximal oropharyngeal colonization by this microorganism [28]. It has also been shown that the maternal immunity conveyed to the fetus during pregnancy diminishes gradually during this period, above all, from 6 to 24 months. Longitudinal investigations have proven that mean immunoglobulin G (IgG) levels at birth are elevated, helping to combat *K. kingae*. Levels then slowly decrease, reaching their lowest point 6–7 months postnatally [31,32]. Nevertheless, IgG levels remain low until 18–24 months old, at which point a progressive increase is noted [31,32]. Infants are, thus, more prone to an OAI due to *K. kingae* between 6 and 48 months old.

Secondly, our results also confirmed that OAIs caused by *K. kingae* presented few acute symptoms and infrequently induced fever. Only 28% of the pediatric cases of *K. kingae* OAI had a temperature >38 °C at admission; indeed, their mean temperature at admission was only 37.4 °C. Even more interesting is the fact that only 17.2% of children with an OAI due to *K. kingae* presented with a temperature above 38.5 °C, a factor that several authors suggest is an essential predictor of an OAI. Analogously, Dubnov-Raz et al. reported how 169 pediatric patients with a culture-proven *K. kingae* OAI frequently presented with a body temperature < 38 °C [31]. Similarly, Chometon et al. reported that only 13 of 39 patients (33.3%) with a *K. kingae* OAI presented with a fever above 38 °C at admission [33]. In three previous studies, we noted that the mean temperatures at the admission of pediatric patients with a *K. kingae* OAI were 37 °C (30 patients) [29], 37.2 °C (66 patients) [10], and 37.3 °C (100 cases) [15], which tends to support those other authors’ findings. Treating physicians, therefore, should bear in mind that most children with an OAI due to *K. kingae* appear in excellent general condition and few of them will present with fever. We can now safely infer that a fever above 38.5 °C is more of an exception than the rule during OAIs due to *K. kingae*. Infants usually present with a symptomatology suggestive of a musculoskeletal disease with either sparing of the impaired limb, a limp, or a refusal to bear weight. Signs on physical examination will usually involve localized pain, limited articular range of motion, or non-traumatic effusion. However, in some cases, it may even prove very difficult to determine exactly where the epicenter of pain is.

The third point to emerge from this study was that *K. kingae* OAIs are characterized by little marked inflammatory biological response, and many treating physicians frequently do not consider this suggestive of an OAI. The mean CRP value was 26.6 mg/L, but CRP levels were less than 20 mg/L in 46.4% of children with a *K. kingae* OAI. The noted increases in CRP values during OAIs due to *K. kingae* remain modest and relatively unspecific. Indeed, it is commonly accepted that viral infections give rise to moderate increases in CRP values, usually up to 25 mg/L. With rheumatoid arthritis, CRP levels can even exceed the 50 mg/L threshold. Although they were surprising, these values matched those published in earlier studies, with CRP levels reaching 37 mg/L for septic arthritis and 18 mg/L for osteomyelitis in a large series of pediatric OAIs caused by *K. kingae* published by Dubnov-Raz et al. [31]. In the same vein, Ilharreborde et al. described a mean CRP level of 39 mg/L in a cohort of 31 children with septic arthritis due to *K. kingae* [12]. In a previous study about OAI due to *K. kingae*, we demonstrated that CRP levels were abnormal (>10 mg/L) in 66% of children (66/100), with a mean level among those 66 of 24 mg/L [15]. This means, therefore, that less than half of patients with an OAI caused by *K. kingae* had a positive value for this predictor, added by Caird et al., for improving the Kocher algorithm’s power to recognize OAIs [25].

Coincidentally, WBC count was considered superior to the normal age-related cut-offs in 16.0% (>17,500/mm^3^ in children 6–12 months old, >17,000/mm^3^ in children 13–24 months old, >14,500/mm^3^ in children 24–48 months old, and >12,000/mm^3^ in older children) of the patients. Furthermore, the WBC count was superior to the described predictor by Kocher of >12,000/mm^3^ [22], only by 52.3%, demonstrating once again that this predictive factor is unreliable for detecting K. kingae OAIs.

Our results also revealed that the value of interpreting ESR is questionable and it must be performed with great caution. Indeed, we noted that, in 75.5% of cases, ESR was higher than 20 mm/h, but its mean value was only 35.3 mm/h. The ESR is a common blood test that can indicate an increase in inflammatory activity within the body caused by one or more conditions, such as autoimmune diseases, infections, or tumors. The ESR is not specific for any one disease, but it is used in combination with other tests to determine the presence of increased inflammatory activity. Infections are usually associated with an extremely elevated ESR (>100 mm/h). The ESRs recorded in the present study are, therefore, only slightly or very rarely suggestive of bacterial infection. In an earlier study, we noted that ESRs were abnormal in 75.6% of the children with a *K. kingae* OAI, with a mean rate of 32.9 mm/h [15]. Similarly, in a study by Dubnov-Raz et al. about infections caused by *K. kingae*, ESRs reached 44.1 mg/L for septic arthritis and 40 mg/L for osteomyelitis [31]. However, when ESRs were interpreted with regard to the cut-off considered as a predictor by Kocher et al. (ESR ≥ 40 mm/h), we noted that only 33.5% of patients with an OAI caused by *K. kingae* had a rate above it. Once more, we had to admit that ESR was a bad predictor of an OAI due to *K. kingae*, even though ESR had previously been considered one of the most sensitive markers of inflammation [34].

Finally, the present study’s results seem to show that the platelet count is probably the best parameter for detecting and recognizing an OAI due to *K. kingae*. Our results underlined that 53.7% of OAIs due to *K. kingae* were characterized by an increase in the platelet count, with a mean value reaching 420,000/mm^3^. Above all, however, 64.2% of the platelet count values of patients with an OAI caused by *K. kingae* were considered to be suggestive when the already validated threshold limit (>361,500/mm^3^) was applied [15]. Indeed, in this study, Coulin et al. have been able to show a statistical difference (*p* < 0.001) of platelet count in the population with *S. aureus* and *K. kingae* OAIs.

It is important to remember that platelets are small anucleate cells, highly specialized for hemostasis and vascular wall repair. Apart from their primary role, it is currently recognized that platelets contribute greatly to systemic inflammatory and immune processes. Indeed, platelet specialization may be part of an evolutionary adaptation that can increase the hosts’ defense against pathogens. Among the extended repertoire of platelet functions, they can secrete pleiotropic immune and inflammatory mediators, factors that play crucial roles in the interactions with monocytes and neutrophils [35].

Thus, thrombocytosis may be reactive and secondary to an underlying inflammatory condition, such as tissue damage, malignancy, or infectious diseases [35]. The definition of thrombocytosis is subject to discussion and varies between a platelet count >400,000/mm^3^ or >500,000/mm^3^. Nevertheless, according to current WHO criteria, a platelet count greater than 400,000/mm^3^ is one of the major criteria for essential thrombocytosis [36]. Thrombopoiesis is usually inhibited after an acute bacterial infection, whereas, in contrast, chronic inflammation is often associated with reactive thrombocytosis [35,37,38]. Until this work, an elevated platelet count was not recognized as a diagnostic marker of an OAI. Clinical experience has taught us that most OAIs due to *K. kingae* present with an increased platelet count, even in the absence of the classic signs of infection, such as a fever or an elevated WBC count [15]. Surprisingly, that reactive thrombocytosis was more often associated with *K. kingae* OAIs than with MSSA infections [15], and this phenomenon is probably explained by the fact that OAIs caused by *K. kingae* generally follow a mild clinical course, probably behaving like a long-standing infection, which can lead to a more sustained reactive thrombocytosis.

However, thrombocytosis may be absent in children with an OAI due to *K. kingae*, and some patients may even present with thrombocytopenia. From a pathophysiological viewpoint, it is now clearly established that interactions with viral infections can occur during an OAI due to *K. kingae*, since damage to the mucosal layer caused by viral agents is believed to trigger *K. kingae* invasive disease by facilitating the bacteria’s spread into the bloodstream and dissemination to distant sites [39]. It is also known that mild thrombocytopenia, often combined with lymphocytopenia, is typical of most of these viral infections, which can thus inhibit the reactive thrombocytosis that is normally encountered during invasive *K. kingae* infections. Thrombocytosis during a viral upper respiratory tract infection in children with musculoskeletal symptoms is all the more significant and can be considered a good predictive marker of an OAI caused by *K. kingae.*

This study has some limitations. First, the fact that OAI due to *K. kingae* occurs almost exclusively in the 6–48-month-old pediatric population due to transitional immunity makes comparison with an older pediatric population with mature immunity and subject to other pathogens impossible. Thus, the present study is based on algorithms and values previously described in the literature.

## 5. Conclusions

Osteoarticular infections due to *Kingella kingae* frequently remain undetected using the classic biological parameters for investigating bacterial infections; the various established algorithms for differentiating musculoskeletal infections from non-septic orthopedic conditions fail to find them. Alongside the standard predictors (C°, WBC, CRP, and ESR), the present study made the serendipitous finding that thrombocytosis is frequently present during OAIs caused by *K. kingae*. Although this biological characteristic is inconstant, it is more frequently present when an invasive infection occurs after a viral infection of the upper respiratory tract—infections that are broadly recognized to induce thrombocytopenia.

## Figures and Tables

**Table 1 microorganisms-11-02130-t001:** Infection type frequencies.

Diagnostic	Counts	% of Total
Septic arthritis	136	55.1%
Osteomyelitis	42	17.0%
Septic arthritis with concomitant osteomyelits	24	9.7%
Spondylodiscitis	16	6.5%
Tenosynovitis	11	4.5%
Septic chondritis	8	3.2%
Myositis	3	1.2%
Transphyseal osteomyelitis	3	1.2%
Sacroiliitis	1	0.4%
Cellulitis/fasciitis	1	0.4%
Abscess	1	0.4%
Pandiaphysitis	1	0.4%

**Table 2 microorganisms-11-02130-t002:** Location frequencies.

Location	Counts	% of Total
Hip	59	27.3%
Knee	48	22.2%
Foot	22	10.2%
Wrist	18	8.3%
Spine	16	7.4%
Ankle	14	6.5%
Hand	13	6.0%
Elbow	9	4.2%
Shoulder	5	2.4%
Sternoclavicular joint	2	0.9%
Tibia	2	0.9%
Femur	2	0.9%
Collarbone	1	0.5%
Distal femur	1	0.5%
Greater trochanter	1	0.5%
Ilium	1	0.5%
Thigh	1	0.5%
Manubriosternal joint	1	0.5%

**Table 3 microorganisms-11-02130-t003:** Distributions of clinical and laboratory parameters.

	Temperature (°C)	WBC (G/L)	Left Shift (%)	PLT (G/L)	CRP (mg/L)	ESR (mm/h)
N	233	243	176	246	237	155
Missing	14	4	71	1	10	92
Mean	37.4	12.7	0.921	419	26.6	35
Median	37.2	12.5	0	405	17	33
Standard deviation	0.897	4.18	2.23	123	27.8	18.9
Minimum	35.8	3.8	0	123	0	2
Maximum	39.8	27.5	24.5	734	165	102

WBC = white blood cell; PLT = platelet count; CRP = C-reactive protein; ESR = erythrocyte sedimentation rate.

**Table 4 microorganisms-11-02130-t004:** Distribution of pathological clinical and laboratory parameters.

	Value	Counts	% of Total
**°C at admission** (n = 231)	>38 °C >38.5 °C	65 39	28.1% 16.9%
**WBC** (n = 241)	>17 G/L >12 G/L	33 127	13.7% 52.7%
**CRP** (n = 235)	>10 mg/L >20 mg/L	152 110	64.7% 46.8%
**ESR** (n = 153)	>20 mm/h >40 mm/h	116 52	75.8% 34.0%
**PLT** (n = 244)	>392 G/L	131	53.7%

WBC = white blood cell; PLT = platelet count; CRP = C-reactive protein; ESR = erythrocyte sedimentation rate.

## Data Availability

Data are available upon reasonable request.

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
