# Peer review of "Biological Predictors of Osteoarticular Infection Due to K. kingae—A Retrospective Cohort Study of 247 Cases"

_microorganisms, 2023, doi:10.3390/microorganisms11092130_

Round 1

Reviewer 1 Report (Previous Reviewer 2)

I have read the resubmitted manuscript titled "Biological predictors of osteoarticular infection due to K. kingae: A descriptive study of 247 cases" with great interest. While some of my previous concerns have been addressed in the new version of the manuscript, there is still a major concern that significantly affects the validity of this study.

The data on inflammatory markers were assessed and categorized based on previous reports and published algorithms. However, in order to assess the predictive ability of these clinical, inflammatory, and hematological markers, it is crucial to compare the studied cohort with control groups of septic pyogenic OAI caused by other pyogenic pathogens, as well as non-septic orthopedic conditions. This is the only way to generate reliable and useful data for clinical applications.

Minor editing may be required.

Author Response

Reviewer 2 Report (New Reviewer)

Dear authors,

In this original paper you sought to to assess the clinical features and biological behavior relating to K. kingae OAI and you also defined the role of platelet count.

Overall, the paper reads well, and no major issues were noted.

I would like you clarify why no formal writing guidelines were followed in this paper (eg the ones found on Equator Network website). Please note adherence to guidelines improves transparency and credibility in reporting.

Please find specific comments below:

Title: The title of this article needs to describe that this work was retrospective in nature. Consider revising.

Line 58: Please note pathogens are not responsible for infections as this verb only refers to human beings. 

Table 1. Please explain why you felt that you should distinguish the clinical entities of septic OA vs septic arthritis.

Line 234: Please replace the word ‘generate’ with a more appropriate one.

-

Round 2

Reviewer 1 Report (Previous Reviewer 2)

I have no additional comments.

This manuscript is a resubmission of an earlier submission. The following is a list of the peer review reports and author responses from that submission.

Round 1

Reviewer 1 Report

In this paper the Authors evaluated some basic tests to search the best one for helping in the diagnosis of  pediatric osteoarticulat infection due to K. Kingae. They suggest that thrombocytosis (?) can be a good marker of such specific osteoarticolar infection.

The idea seems difficult to be shared: all the infections, expecially in children, are commonly associated to high platelet counts as well with increased ESR, CRP and other test of inflammtion

Moreover:

1) The statistical evaluation of the data is poor and the absence of a patients  control group reduces the data reported to a mere light information. The paper could be greatly improved comparing the laboratory results obtained in this setting and to other settings of children.

2) As the authors stated, thrombocytosis is usually consider when the platelet count is above the normal range (150-450 x 10^9/L ) and in children the lower limit to consider thrombocytosis is 500 x 10^9/L  in repeated controls. How can the author consider "increased" a platelet count of 361, 5 x 10^9/L)?

3) Results: line 161: which are the cited exclusion criteria?

On a whole, I think that the paper need a profound restiling

Reviewer 2 Report

In this study, the authors investigated the clinical presentation and biological inflammatory response to K. kingae OAIs in a large cohort of pediatric patients. They revealed that thrombocytosis was frequently present during OAIs caused by K. kingae. I have several concerns that need to be addressed, as follows:

  • The title should be revised to reflect the objectives of the clinical and biological inflammatory response to K. kingae OAIs, not only thrombocytosis.
  • Methods: The methods used for estimating WBC count, platelet counts, ESR, and serum CRP should be described in detail.
  • Results:

-        It is feasible to illustrate the screened, excluded, and included patients in a flow chart.

-        The main concern in this study is the lack of comparative groups (septic pyogenic OAI caused by other pyogenic pathogens and non-septic orthopedic conditions) to compare the clinical and inflammatory responses between these groups and those with K. kingae OAIs.

  • Discussion: There are several limitations to this study that should be acknowledged as the data (inflammatory markers) were assessed and categorized based on previous reports and published algorithms.
  • The manuscript should be edited for some structural and grammatical errors.
  • The manuscript should be edited for some structural and grammatical errors.